# Evaluating the Relationship Between the Introduction of the Acellular Pertussis Vaccine and Whooping Cough Resurgence in the United States

**DOI:** 10.3390/vaccines13080841

**Published:** 2025-08-07

**Authors:** Jeegan Parikh, Ismael Hoare, Ricardo Izurieta

**Affiliations:** 1College of Public Health, University of South Florida, Tampa, FL 33612, USA; jeeganparikh@usf.edu; 2School of Public Health and Health Sciences, California State University, Dominguez Hills, Carson, CA 90747, USA; rizurieta@csudh.edu

**Keywords:** whooping cough, *Bordetella pertussis*, DTaP, DTP, Tdap, whole-cell vaccine, acellular vaccine, pertussis

## Abstract

**Background/Objectives**: The whole cell pertussis vaccine was introduced in the United States in the 1940s and switched to the acellular pertussis vaccine partially in 1992 and completely in 1997. This study examines the relationship between the resurgence of pertussis in the United States and the change in the type of pertussis vaccines. **Methods**: Pertussis cases from 1922 to 2024 were obtained from the CDC’s national notifiable disease surveillance system, and vaccination coverage was obtained from the WHO. A trend analysis and Pearson’s correlation test were conducted between the incidence of cases and the coverage of the third and fourth doses of the pertussis vaccine. An ANOVA test and multivariable linear regression were performed to assess the association between the type of vaccine and the number of pertussis cases. **Results**: The number of cases increased from 4083 in 1992 to 35,435 in 2024, with cyclical outbreaks in 2010, 2012–2014, and 2024. The third and fourth doses of pertussis vaccine coverage had mild and moderate correlations with the number of pertussis cases. The vaccine type had a significant association with the number of pertussis cases and stayed significant after adjusting for vaccination coverage. **Conclusions**: The switch in pertussis vaccine has impacted the epidemiology of pertussis outbreaks in the United States. A combination of factors, such as different types of immune response to vaccines, waning of immunity, and selection of non-vaccine bacterial strains, may explain the observed results. Further research on newer, improved vaccinations or alternative schedules in children needs to be conducted to address the resurgence of pertussis in this study.

## 1. Introduction

Pertussis, also known as whooping cough, is a highly contagious disease caused by the Gram-negative bacterium *Bordetella pertussis* [1]. The bacterium produces multiple exotoxins that produce the signs and symptoms associated with the disease. These active bacterial products include pertussis toxin, pertactin, tracheal cytotoxin, agglutinogens, and filamentous haemagglutinin and the pathogen is transmitted from person to person by respiratory droplets. It has a secondary attack rate of 11–18, requiring a vaccination coverage of 90–95% for population-level immunity and prevention of outbreaks of disease [2]. Whooping cough can develop in all age groups, but the disease and associated complications are more severe in infants or younger children [1].

Before the availability of the vaccine, pertussis was one of the most common diseases of childhood and childhood mortality in the United States. A whole-cell pertussis vaccine was introduced in the 1940s, and it helped in reducing the number of cases of the disease by 75% [3]. However, whole-cell vaccination had several vaccine-associated adverse events, such as pain, fever, and erythema, leading to refusal from parents to vaccinate their child [4,5]. The complaint from parents regarding the adverse events of the vaccine had resulted in a reduction in vaccination coverage in the second half of the 1980s [4,5,6]. Additionally, the incidents of febrile seizures [7] or acute encephalopathy [8] after whole-cell pertussis vaccination (DTP) led to reluctance among physicians in developed countries to administer the whole-cell pertussis vaccine (DTP). This led to the introduction of the acellular pertussis vaccine (DTaP) with lower adverse events [9]. The introduction of the acellular vaccine helped in overcoming the fears, and vaccination coverage improved among the population, reaching the optimum level by 1994 [6]. The acellular vaccine consists of five purified antigens: pertussis toxin, filamentous hemagglutinin, pertactin, and fimbria types 2 and 4 and was developed to address the concerns about the safety profile of the whole-cell pertussis vaccine [9]. This acellular vaccine replaced the whole-cell vaccines partially as the fourth and fifth doses in 1992 (Figure 1) and altogether in 1997 (Figure 2) [9,10]. The current recommended immunization schedule for children mandates a primary series of acellular pertussis vaccine (DTaP) at two, four, and six months and two booster doses, i.e., the fourth and fifth doses of DTaP at 15–18 and four–six years of age (Figure 3).

In vitro studies showed acellular pertussis vaccine has similar efficacy as whole-cell vaccine, i.e., 60–89%; however, there has been an increase in the number of cases of pertussis in the United States since the introduction of the acellular pertussis vaccine [3,11,12]. This increased incidence despite the immunization coverage of the three doses of acellular pertussis vaccine (DTaP) in the United States around the recommended 90–95% has raised questions on the efficacy of the acellular pertussis vaccine [6]. The incidence of the disease among vaccinated children suggests the waning of immunity could be a potential reason behind the increase in incidence [11,13]. To address the issue of the increase in incidence of disease and potential waning of immunity, a single booster dose of acellular pertussis (Tdap) vaccine was introduced for those aged 11 or 12 years in 2005 (Figure 3). Despite this, only 28.6% of the adults aged 18 years or older had received the Tdap vaccine in 2022, compared to the estimated coverage in 2019 and 2013 of 28.9% [14,15]. In addition, the possibility of lower vaccination coverage in certain groups or population pockets may be due to lack ofaccess to healthcare or impacted by the anti-vaccination movement [2,16].

There was significant disparity seen in coverage of vaccination among Black children and those who were uninsured or covered by Medicaid [16]. Therefore, certain sections among the population had lower levels of vaccination coverage compared to the required vaccination coverage for herd immunity.

The primary goal of this study is to evaluate this resurgence of pertussis cases in the United States, its relationship with vaccination coverage of the pertussis vaccine, and the distinct types of pertussis vaccines. Further, the available data from the National Notifiable Disease Surveillance System and surveillance reports of the Centers for Disease Control and Prevention will be used to make inferences about the epidemiological pattern of pertussis resurgence.

## 2. Methods

This study was conducted to evaluate the relationship between the resurgence of pertussis (whooping cough) cases in the United States and the transition from whole-cell pertussis vaccine (DTP) to acellular pertussis vaccine (DTaP). Specifically, vaccination coverage of three doses of whole-cell vaccine or acellular pertussis vaccine was compared with the annual incidence or annual number of pertussis cases in the U.S.

### 2.1. Study Design

A population-level ecological study design was employed to assess the impact of whole-cell versus acellular pertussis vaccine in the population. The whole-cell vaccine for pertussis was introduced in the 1940s and was gradually replaced by the acellular pertussis vaccine (aP), partially in 1992 and fully in 1997 [10]. The normal immunization schedule of the pertussis vaccine includes three primary doses at two, four, and six months, followed by boosters at 15–18 months (fourth dose) and four–six years (fifth dose) [1]. Given that 90% of the pertussis cases are found in individuals less than 10 years of age, the study focused on individuals aged 1 month to 20 years living in the U.S. in the period of 1980 to 2024.

### 2.2. Data Sources and Statistical Analysis

Pertussis is a national notifiable infectious disease in the United States, and thus, annual case counts and incidence data categorized by age groups were obtained from the National Notifiable Disease Surveillance System (NNDSS) and surveillance reports of the Centers for Disease Control and Prevention (CDC) [3,17]. Vaccination coverage data for the third and fourth doses of whole-cell pertussis (DTP) or acellular pertussis (DTaP) vaccination coverage levels were obtained from the World Health Organization (WHO) databank on immunization coverage in the United States and the CDC [6].

Initially, a trend analysis was conducted to compare the number of cases and vaccination coverage of third and fourth doses of whole-cell and acellular pertussis vaccines, followed by a correlation analysis between vaccination coverage and the annual number of pertussis cases. An ANOVA test was then conducted to examine the differences in mean pertussis cases in the three key periods: before 1992 (whole-cell vaccine era), after introduction of acellular pertussis vaccine (DTaP) but before introduction of Tdap, i.e., 1992–2005, and after introduction of Tdap (2005–2023). This was followed by multivariable linear regression analysis to find the independent effect of vaccine types and coverage on the pertussis cases. Additionally, age-stratified trend analysis of the annual incidence number in the age groups of <1 year, 1–6 years, 7–10 years, 11–19 years, and >20 years was conducted. Graphs for trend analysis were created in Microsoft Excel (2025 ^®^Microsoft Corporation, Redmond, WA, USA), and statistical analysis was performed in RStudio ^®^2025 (Posit Software PBC, Boston, MA, USA).

### 2.3. Ethical Considerations

The study used publicly available de-identified secondary data, thus the permission for use of data for analysis was implied with the acknowledgement of the ownership of the original data. The quality of the data was assessed for accuracy by reviewing the period of data collection and content of data. Furthermore, suitable analysis of the data was conducted to follow the principles of ethical usage of data.

## 3. Results

### 3.1. Pertussis Incidence

The trend analysis of the annual number of cases from 1922 to 2024 shows a sharp decline in the number of cases from 107,473 in 1922 to 35,435 in 2024 (Figure 4). However, a deeper analysis suggests an interesting pattern. Initially, in the period of 1922–1948, the annual number of pertussis cases was above 100,000, with a peak of 265,269 in 1934. This was followed by a decline in cases from 120,718 in 1950 to 1730 in 1980. This decline coincides with the introduction of the whole-cell vaccine in the 1940s. The lowest number of cases was seen in 1981, with only 1248 cases reported. There were also periodic outbreaks of pertussis reported in 2010, 2012–2014, and 2024, when cases reported were above 30,000 per year. The trend analysis suggests that the continuation of this pattern should have resulted in the elimination of the disease in 2002–2004, but there was a resurgence of the disease in the 1990s with the increase in the incidence of pertussis, reaching a peak of 48,277 in 2014. Another interesting pattern to be noticed is the sharp fall in pertussis cases during and immediately after the COVID-19 pandemic, followed by a sharp rise post-2022.

An interesting epidemiological trend was noticed when age-stratified incidence of pertussis cases was analyzed (Figure 5). The incidence of pertussis was highest in the <1-year-old age group (Figure 5). This was followed by the 1–6, 7–10, 11–19, and 20+ year age groups in ascending order. A steady increase in the incidence was noted in the 7–19 years of age group until the pandemic, i.e., 2020. This was reflected in the outbreak years of 2010–2014, when the 7–10-year-old age group had the second-highest incidence, followed by the 11–19-, 1–6-, and 20+-year-old age groups (Figure 5). The incidence per 100,000 population in the less than 1-year-of-age group increased from 38.41 in 1992 to 96.47 in 2005 and 59 in 2019 and then decreased to 23.16 in 2023 (Table 1). In the age groups of 7–10 years and 11–19 years, the incidence per 100,000 cases was 4.52 and 3.75 in 2023 and 15.05 and 15.01 in 2019, compared to 1.95 and 1.99 in 1992, respectively (Table 1). During and immediately after the COVID-19 pandemic, a significant dip in pertussis cases was observed across all age groups, followed by a rise in cases in all age groups except those 20+ years from 2022 onwards. In the outbreak years of 2010, 2012, 2013, and 2014, the incidence was as high as 31.78, 58.52, 30.61, and 34.04 in the 7–10-year-old age group and 13.3, 38.02, 21.27, and 29.57 in the 11–19-year-old age group (Table 1). The incidence in the age groups of 1.6 years and 20+ years increased from 3.93 and 0.27 in 1992 to 16.31 and 1.78 in 2019 and 9.22 and 0.76 in 2023 (Table 1).

### 3.2. Pertussis Vaccine Coverage

Similar analysis of DTP/DTaP third dose coverage in the population showed consistent 90–92% coverage except for a sudden decline in the late eighties, with coverage being as low as 69% in 1991 (Figure 6). For the DTP/DTaP fourth dose, a consistent 80–82% coverage was seen among the population (Figure 7). No decrease in the vaccination coverage of the DTaP third and fourth doses was noticed during and immediately after the COVID-19 pandemic (Figure 6 and Figure 7).

Comparing the annual number of cases vs. DTP/DTaP-3 coverage, the initial decrease in coverage was correlated with an increase in the number of cases of pertussis. However, despite the increase in the vaccination coverage, the incidence of pertussis did not decrease back to levels seen in the 1980s. Pearson’s correlation between number of cases and DTP/DTaP 3 or DTP/DTaP 4 was 0.1 (*p* = 0.5) and 0.36 (*p* = 0.04), respectively. A significant association was found between the vaccine types and number of pertussis cases (F = 11.75, *p*-value < 0.001). A multivariable linear regression showed that the type of vaccine was significantly associated with the number of cases even after adjusting for third or fourth vaccine dose coverage (*p* < 0.05). After adjusting for vaccination coverage, on average 7206 more cases of pertussis were seen after the switch from whole-cell vaccine to acellular pertussis vaccine in 1992, while after 2005, on average 16,096 more cases of pertussis were seen (Table 2). The model explained 30–32.5% of the variance in the pertussis cases in the U.S. and did not violate the assumptions of linear regression, i.e., independence of independent variables and homoscedasticity of residuals (Figure 8).

## 4. Discussion

The results show that the trend of the incidence of pertussis disease has reversed since the 1990s, despite the vaccination coverage of the third dose of pertussis vaccine being about the required 90–95% in the population. Initially, the reversal of the trend in pertussis incidence was suggested to be due to better surveillance and diagnostic techniques [18], but it does not explain the exponential growth in cases with periodic outbreaks or the sudden dip during the COVID-19 pandemic. Despite the high vaccination coverage of the third dose of the pertussis vaccine, the coverage of the fourth booster dose of the pertussis vaccine has plateaued and remains around 82–85%. A significant moderate correlation was observed between the vaccination coverage of the fourth dose of the pertussis vaccine and the incidence of pertussis cases. These findings suggest a need to educate the parents about the importance of booster doses, i.e., the fourth and fifth doses of the pertussis vaccine, in the prevention of pertussis.

The drop in pertussis cases observed during and immediately after the COVID-19 pandemic could be attributed to COVID-19 mitigation factors such as wearing face masks and social distancing requirements [19]. There was no significant decrease in vaccination coverage of the third or fourth doses of pertussis in the United States during and after the pandemic. The reduction in cases during the pandemic, followed by a return to pre-pandemic levels by 2024 after the cessation of COVID-19 mitigation measures, supports the hypothesis that this decline was primarily due to social distancing and face masks. This pattern is similar to the epidemiology of pertussis seen in Europe, as well as middle- and low-income countries [19,20].

The type of vaccine was significantly associated with the incidence of pertussis cases. This association remained significant even after adjusting for the third or fourth doses of vaccination coverage. In 2015, Gambhir et al. proved that acellular pertussis was associated with higher waning of immunity than the whole-cell pertussis vaccine [21]. The whole-cell vaccine series has a rate of loss of immunity at the same rate as natural infection, i.e., 3 × 10^−5^ yr^−1^ and results in lifelong immunity. The acellular pertussis vaccine results in the loss of immunity at a much higher rate of 0.018 yr^−1^ [21]. Other studies have suggested that the immunity after the acellular pertussis vaccine starts to wane after 2 years, compared to 7 years with the whole-cell vaccine or natural infection. Additionally, acellular pertussis vaccine leads to production of a Th2-polarized immune response, while whole-cell vaccine leads to a Th1/Th17 immune response [22,23]. The Th2-polarized immune response results in the persistence of *B. pertussis* in biofilms, while the Th1/Th17 immune response leads to its clearance [23]. Therefore, the acellular pertussis vaccine fails to prevent the infection or transmission of *B. pertussis* compared to the whole-cell pertussis vaccine [24,25]. The waning of immune response along with the type of immune response to the acellular pertussis vaccine and its effect on infection transmission could be one of the reasons behind the trend seen in the incidence of pertussis in the United States. Furthermore, the waning of immunity may have resulted in the immunological selective pressure leading to the emergence of non-vaccine antigen strains of the pertussis bacterium [26,27]. This has resulted in a change in the antigenic and genotypic features of the circulating pathogenic *B. pertussis* strains without the two vaccine antigens: pertactin and pertussis toxin [28,29,30,31]. The emergence of strains without vaccine antigens and the ability to cause disease among vaccinated populations have raised further doubts over the effectiveness of the acellular vaccines [28,29,31].

The highest incidence of pertussis has been seen in the age group of less than 1 year of age. This higher incidence in infants could be attributed to the lack of maternal Tdap vaccination, which may be due to limited access to healthcare, vaccine hesitancy, and other factors. The Tdap vaccination coverage among pregnant women has stayed consistent around 52.8% (2019)–55.4% (2023), putting nearly 50% of the infants at risk of pertussis infection [32,33]. The lack of Tdap vaccination among pregnant women leads to a lack of transfer of maternal pertussis antibodies to newborns and infants and thus a higher incidence among infants. This hypothesis is further supported by the highest incidence among the infants being seen in infants of three months of age [34]. Pre-pandemic (2020), there was a change in the epidemiological profile of the disease, with nearly one-third of cases seen among the 7–19-year-old age group. The lack of change of the trend despite the introduction of the acellular pertussis vaccine (Tdap) booster dose and incidence of cases among vaccinated individuals supports the hypothesis of waning of immunity and the emergence of a *B. pertussis* strain with non-vaccine antigens being responsible for the increase in pertussis cases and cyclical outbreaks [26,27].

The result of our study shows the significant impact of vaccine type on the resurgence and epidemiology of the disease in the United States. Similarly, a resurgence of the disease was seen in Brazil and Mexico [35]. Zerbo et al. (2022) suggested that modification of the vaccination schedule and new vaccination strategies are needed to counter the waning of immunity and suboptimal vaccine effectiveness and reduce the prevalence of the disease [36]. Research studies have shown that children who were primed with acellular pertussis vaccine in 1997 had a much higher rate of pertussis cases than those who were primed with a whole-cell vaccine between 1992 and 1997 [37,38]. An alternative schedule of priming with the first three doses with a whole-cell vaccine and then a booster with an acellular pertussis vaccine can be considered to address the issue of waning of immunity, different types of immune responses, and emergence of strains of bacteria that lack vaccine antigens. The recommendations for Pertussis vaccinations should be made considering the balance between vaccine side effects and vaccine effectiveness. The current recommendations of complete immunization with acellular pertussis vaccine and specifically increasing the coverage of booster doses should be continued until further effective vaccines are developed or alternative schedules are recommended by the Advisory Committee on Immunization Practices.

This study has certain limitations because of its ecological study design. The data used in the study is group-level data and thus cannot find the temporal association between the acellular pertussis vaccine and incidence of pertussis. The ecological study design also means there is a lack of individual data linking the case of pertussis in children less than one year to the type and time of maternal vaccination. Further, there is a lack of Tdap vaccination coverage in the analysis. This is primarily because of methodological reasons, as the Tdap booster was introduced in 2005, and there is no directly comparable data for the whole-cell pertussis vaccine. Additionally, most of the available data related to Tdap coverage pertains to administration during pregnancy. However, to address this limitation, we have divided the vaccine type into three periods to reflect the period before and after the introduction of the Tdap booster dose. This study also does not consider pockets of the unvaccinated population, which could be inflating the number of pertussis cases in the population.

## 5. Conclusions

The resurgence of a disease once thought to be under control reflects a critical challenge for the public health community and highlights the need for renewed attention and coordinated action. The resurgence of pertussis suggests the need for continuous environmental surveillance to understand the effects of immunization on the genotype of bacteria. Further immunological and microbiological studies are required to explain the association between the acellular pertussis vaccine, the emergence of new strains of bacteria, and the increase in the number of pertussis cases. In the future, we would like to conduct a cross-sectional study comparing the incidence of pertussis in children less than one year of age to the type and time of maternal vaccination. In addition to further research, co-operation between different agencies and leadership is necessary to gain control and prevent the cyclical outbreaks of pertussis.

## Figures and Tables

**Figure 1 vaccines-13-00841-f001:**
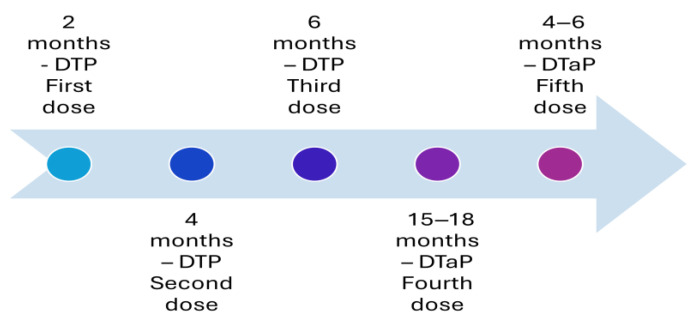
Recommended immunization with vaccines with an acellular pertussis component in the United States in 1992.

**Figure 2 vaccines-13-00841-f002:**
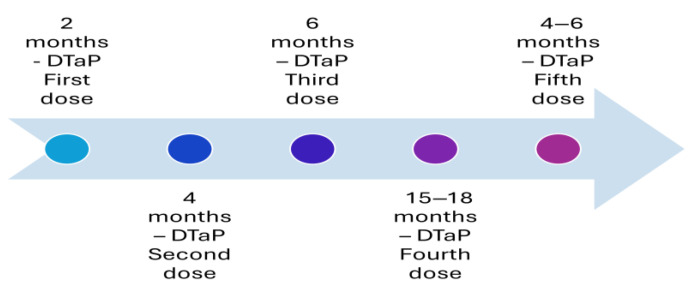
Recommended immunization with vaccines with an acellular pertussis component in the United States in 1997.

**Figure 3 vaccines-13-00841-f003:**
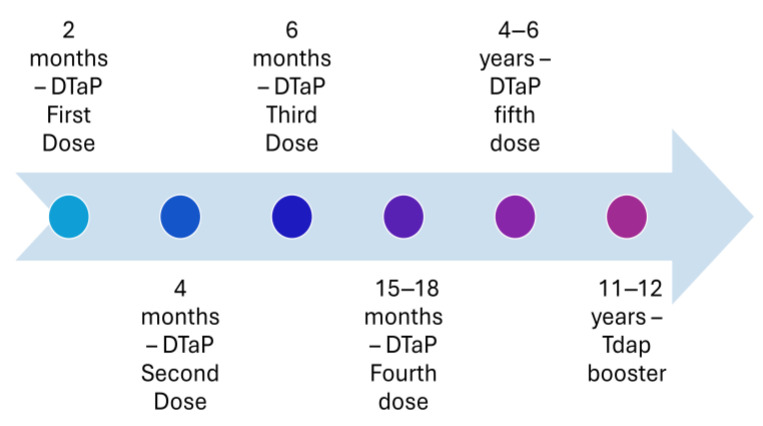
Recommended immunization with vaccines with an acellular pertussis component in the United States in 2005.

**Figure 4 vaccines-13-00841-f004:**
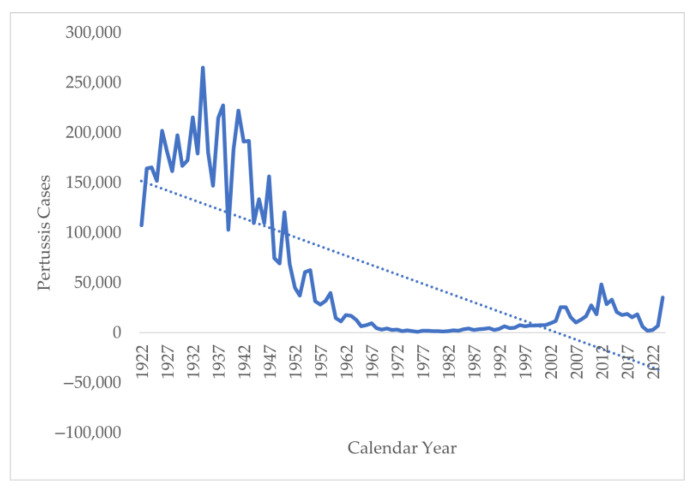
Number of pertussis cases in the United States from 1922 to 2024.

**Figure 5 vaccines-13-00841-f005:**
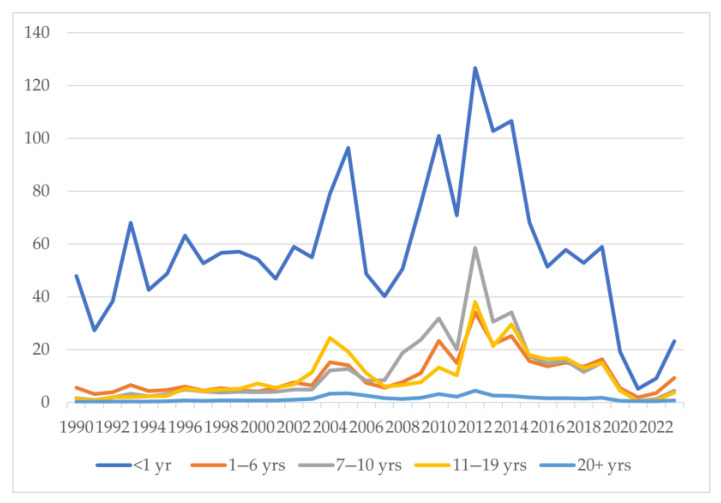
Incidence of pertussis per 100,000 people by age groups in the United States.

**Figure 6 vaccines-13-00841-f006:**
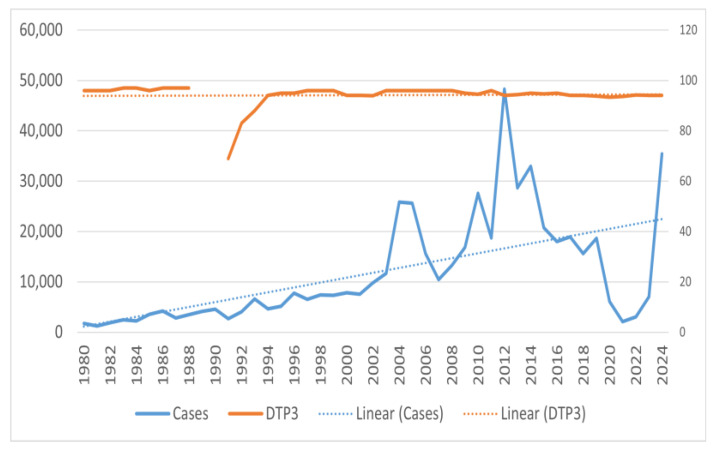
Number of pertussis coverage vs. DTaP third dose coverage in the United States.

**Figure 7 vaccines-13-00841-f007:**
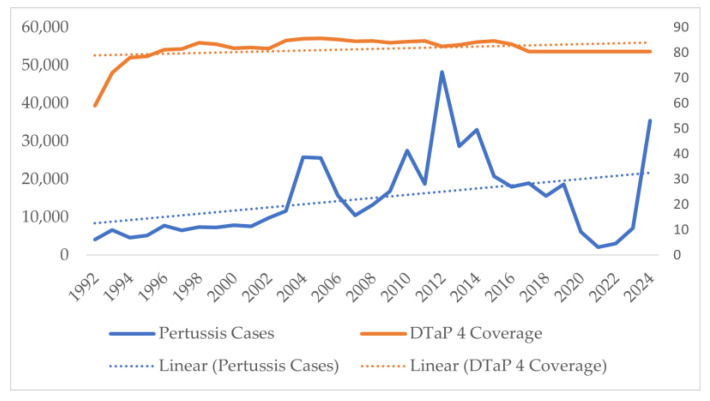
Number of pertussis coverage vs. DTaP fourth dose coverage in the United States.

**Figure 8 vaccines-13-00841-f008:**
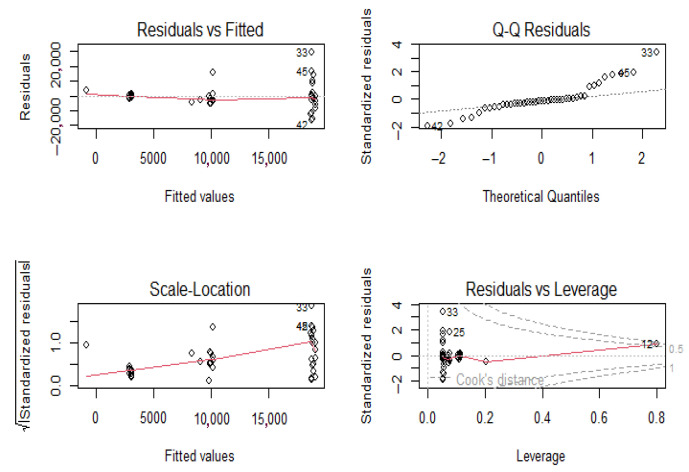
Homoscedasticity of residuals.

**Table 1 vaccines-13-00841-t001:** Incidence of pertussis per 100,000 people by age group.

Year	<1 yr	1–6 yrs	7–10 yrs	11–19 yrs	20+ yrs
1990	47.89	5.57	1.68	1.38	0.19
1991	27.27	3.2	0.96	0.58	0.14
1992	38.41	3.93	1.95	1.99	0.27
1993	68	6.59	3.26	2.08	0.33
1994	42.61	4.35	2.26	2.3	0.27
1995	48.74	4.71	3.17	2.41	0.42
1996	63.23	6.07	4.81	5.25	0.72
1997	52.73	4.49	4.23	4.33	0.66
1998	56.71	5.45	3.8	4.88	0.78
1999	57.12	4.57	3.98	5.17	0.74
2000	54.23	4.15	3.9	7.08	0.74
2001	46.84	5.49	4.09	5.58	0.79
2002	59.01	7.53	4.94	6.8	1.08
2003	54.98	6.41	4.81	11.34	1.32
2004	79	15.28	12.07	24.5	3.29
2005	96.47	14.15	12.64	19.17	3.5
2006	48.78	7.47	8.32	11.05	2.63
2007	40.25	5.64	8.48	6.02	1.57
2008	50.54	7.67	18.85	6.58	1.35
2009	75.23	11.16	23.76	7.75	1.73
2010	100.9	23.27	31.78	13.3	3.15
2011	70.89	15.02	20.05	10.26	2.15
2012	126.65	34.09	58.52	38.02	4.51
2013	102.77	22.09	30.61	21.27	2.61
2014	106.68	25.14	34.04	29.57	2.5
2015	68.1	15.6	17.45	17.9	1.9
2016	51.41	13.65	14.84	16.31	1.68
2017	57.78	15.16	15.79	16.83	1.68
2018	52.8	13.5	11.6	13	1.4
2019	59	16.31	15.05	15.01	1.78
2020	19.25	5.39	4.7	4.46	0.68
2021	5.22	1.94	0.46	0.45	0.49
2022	9.07	3.64	1.3	0.79	0.54
2023	23.16	9.22	4.52	3.75	0.76

**Table 2 vaccines-13-00841-t002:** Association between vaccine type and pertussis cases adjusting for vaccination coverage.

Vaccine Types	Beta Estimate	*p*-Value
Whole-Cell Vaccine	Reference	
Acellular Pertussis Vaccine (1992–2005)	7206.6	0.058
Acellular Pertussis Vaccine with Tdap Booster (2005–2024)	16,096.2	0.00001

## Data Availability

This study uses publicly available de-identified secondary data from CDC and WHO for analysis and interpretation. The link to the data has been cited in the reference list for data availability.

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
