# Peer review of "Evaluating the Relationship Between the Introduction of the Acellular Pertussis Vaccine and Whooping Cough Resurgence in the United States"

_vaccines, 2025, doi:10.3390/vaccines13080841_

Round 1
Reviewer 1 Report
Comments and Suggestions for Authors
This paper documents further a pint that has been underlined in many publications: the resurgence of pertussis after switch from whole cell pertussis vaccine to acellular pertussis vaccine. The basis of this assessment is an ecological analysis comparing vaccine coverage and pertussis incidence in the USA. Possible explanations were obtained by adequate analysis of the literature.
Minor comments
- Point 3.1 and line 216 of discussion. I am surprised that no comment is included about lower incidence in the COVID-19 pandemic years. The only allusion I see is at line 216. More comment is needed.
- Point 3.2. You comment about vaccine coverage in the late eighties, but figure 3 starts only at year 1992.
- Multiple places: please respect the typographical conventions: Bordetella pertussis has to be written in italic, with first letter uppercase for the genus and lower case for the species; whopping cough and pertussis without uppercase, …
- Line 42-43: in addition to adverse events that affect vaccine acceptance by parents, I suggest mentioning also more severe adverse events that affect vaccine acceptance by physicians, in particular inconsolable crying and most feared, acute encephalopathy. For this reason, reintroduction of whole-cell vaccine is probably impossible in developed countries.
- Reference 12: where can this be accessed? Is there a website?
Author Response
- Point 3.1 and line 216 of discussion. I am surprised that no comment is included about lower incidence in the COVID-19 pandemic years. The only allusion I see is at line 216. More comment is needed.
Thank you for this feedback. We have addressed this by specifically including information related to the lower incidence or number of pertussis cases in the COVID-19 pandemic years. In results, lines 150-151 and line 164-167 under 3.1, line 180-182 under 3.2 has been added to reflect this dip in cases during pandemic years. Further, lines 217-225 have been included in the revised manuscript in discussion to explain the lower incidence of pertussis during COVID-19 pandemic.
- Point 3.2. You comment about vaccine coverage in the late eighties, but figure 3 starts only at year 1992.
The figure has been changed to include the data from 1980. This is primarily because data on vaccine coverage for third dose is only available from 1980s. Data for fourth dose of vaccine coverage is not available before 1992.
- Multiple places: please respect the typographical conventions: Bordetella pertussis has to be written in italic, with first letter uppercase for the genus and lower case for the species; whopping cough and pertussis without uppercase, …
Thank you for pointing this out. This has been changed in manuscript to respect the typographical conventions.
- Line 42-43: in addition to adverse events that affect vaccine acceptance by parents, I suggest mentioning also more severe adverse events that affect vaccine acceptance by physicians, in particular inconsolable crying and most feared, acute encephalopathy. For this reason, reintroduction of whole-cell vaccine is probably impossible in developed countries.
Thank you for this information. We have added information on febrile seizures and encephalopathy after whole cell vaccine with relevant references (line 47-50) in the revised manuscript.
- Reference 12: where can this be accessed? Is there a website?
The reference has been modified to include a link to the webpage for accessing the reference. Because of the other revisions, reference 12 is reference 14 in the revised manuscript.
Reviewer 2 Report
Comments and Suggestions for Authors
Dear Authors!
Thank you for the opprotunity to read and review your manuscript!
The pertussis is an important infection, especially in young children less than 1 year
The Authors proviided the anaylysis how the change of the vaccine type associted with pertussis outbreaks and made the necceassary analysis with booster vaccination adjustment
The intorduciton contains the information about the actuality
The Methods were decribed in details
The statistical analysis was relevant to study's aims
The results are clear and very intersting from the practical point of viewe. The information about the maternal antidodies and vaccination during the pregnancy might improve the sitaution with pertussis prevalence in young children less than 1 year old
The discussion contains the rtelevant literature, the Authors compared their data and do the neccesary points for the imrovement of the pertussis incidence
The Discussion has a Limitation section there Authors discloused the weak parts of their study
The conclusion supports the main study results
The manuscript has tables and fugures making the results more clear
I can only might reccomend to provide a figure with different types of vaccination in different years, including the additional dose since 2005 to better understand the vaccination process
Is it possible to compared the incidence of the pertussis in children younger 1 year with the type of maternal vaccination and time since the last maternal vaccination, e.g. younger mothers could be vaccinated with new vaccine, but the time since the last vaccination also was shorter. This poitnwill be very interesting from the practical point if view
Author Response
- I can only might reccomend to provide a figure with different types of vaccination in different years, including the additional dose since 2005 to better understand the vaccination process
Thank you for this feedback. The figures will help to better understand the change in vaccination process throughout the years. Three different figures i.e. Figure 1, figure 2, figure 3 have been added to the revised manuscript to reflect the timeline or changes in types of pertussis vaccination in 1992, 1997 and 2005.
- Is it possible to compared the incidence of the pertussis in children younger 1 year with the type of maternal vaccination and time since the last maternal vaccination, e.g. younger mothers could be vaccinated with new vaccine, but the time since the last vaccination also was shorter. This poitnwill be very interesting from the practical point if view
Because of the limitation of the study design and lack of specific data, we could not implement this request. We have included this in the manuscript as a limitation of the study design (lines 284-286). Also, considering the potential practical implications, this is something we would like to conduct in future and thus have included under the conclusion section of the manuscript (lines 302-304).